# Genetic Diversity and Population Structure in Farmed and Wild Pacific Oysters (*Crassostrea gigas*): A Comparative Study

**DOI:** 10.3390/ijms26094172

**Published:** 2025-04-28

**Authors:** Kang-Rae Kim, Jong-Won Park, Kyung-Il Park, Hee-Jung Lee

**Affiliations:** 1Southeast Sea Fisheries Research Institute, National Institute of Fisheries Science, Namhae 52440, Republic of Korea; kimkangrae9586@gmail.com; 2Southeast Sea Fisheries Research Institute, National Institute of Fisheries Science, Tongyeong 53085, Republic of Korea; dapowind@korea.kr; 3Kunsan National University, Gunsan 541150, Republic of Korea; kipark@kunsan.ac.kr

**Keywords:** microsatellite, effective population size, aquatic animals, shellfish

## Abstract

The Pacific oyster, *Crassostrea gigas*, is an important commercially farmed species in Korea. *C. gigas* exhibits low genetic diversity in wild populations in Korea. To address this, we bred Japanese broodstock for more than five generations and released them into two populations to increase genetic diversity. We also assessed whether this improvement was achieved by comparing them with a control population. In this study, we analyzed genetic diversity using 16 microsatellite loci of *C. gigas*. The observed heterozygosity *H*_O_ in the farmed population ranged up to 0.494, while in the wild population, it was 0.437. The farmed population had the highest genetic diversity, but the effective population size was low (105). The PD population size for resource creation was 403, which was higher than that of GH. The genetic structure was divided into two groups with *K* = 2. The first group consisted of the BR population, while the second group included the GH, GW, and PD populations. Therefore, we confirmed significant genetic differences between the farmed, wild, and resource creation populations. This study provides essential genetic information for future fishery resource development and conservation of *C. gigas*.

## 1. Introduction

Wild Pacific oysters (*Crassostrea gigas*; An et al., 2013, Korea; Li et al., 2006, China; Zhang et al., 2023, China) from China and Korea are reported to have low genetic diversity [1,2,3]. The genetic diversity of Pacific oysters inhabiting Europe is reported to be higher than that of the Chinese and Korean populations [4]. Genetic diversity is the driving force for species to adapt to various habitats [5,6]. Pacific oysters in Korea are often reported to have low genetic diversity due to aquaculture [3]. Maintaining genetic diversity in Pacific oyster aquaculture is essential to prevent population decline due to diseases and disasters [3,4,5,6]. In particular, methods to increase genetic diversity are being sought due to the decrease in genetic diversity in aquaculture populations [3]. One way to increase genetic diversity is through reinforcement by releasing individuals from a population with high genetic diversity [7]. These attempts have been effective and are considered one of the most important methods.

Another way to prevent the decrease in genetic diversity is to increase the effective population size [8]. It is suggested that the effective population size should be 100 or more in the short term and 1000 or more in the long term [3]. This is because the lower the effective population size, the higher the rate of homozygosity over generations, which reduces heterozygosity and, consequently, genetic diversity [3,4,5,6]. Therefore, maintaining a high effective population size in Pacific oyster farming is an important factor in resource creation. In fact, a previous study reported that when a population was created for the purpose of resource creation through multiple populations from different regions, the effective population size became effectively infinite, leading to the formation of a healthy population [8]. Since Pacific oysters are also reported to have low genetic diversity [3], attempts to reinforce them using the same method as in the previous study are needed to increase genetic diversity and effective population size [8].

Japanese Pacific oysters were found to have relatively high genetic diversity compared to Korean Pacific oysters [3,9]. When the genetic diversity of other populations is high like this, it can be utilized to increase genetic diversity for resource creation purposes [8]. In the case of a population for resource creation, it is important to secure a population with as many alleles and as much genetic diversity as possible [10]. However, it is often difficult to secure and prepare such broodstock [11]. Therefore, to achieve the maximum effect at a limited cost, we need a population with as much diversity as possible [12,13].

Differences in genetic structure in aquaculture are one of the factors to consider when using the reinforcement method [8]. This is because when there is such a difference in genetic structure, outbreeding may actually reduce genetic diversity [8,14,15,16]. Therefore, it appears important to create resources by focusing on a population with high genetic diversity among populations with little difference in genetic structure [14,15,16].

Although previous studies have addressed the issue of reduced genetic diversity in *C. gigas*, comprehensive comparative studies of genetic diversity between wild and farmed populations have been lacking, and we aimed to investigate this issue. Our study is expected to serve as an important genetic indicator on the effects of releasing broodstock for the purpose of resource creation for wild populations. Additionally, experiments were conducted on the hypothesis that there would be genetic differences between Japanese populations bred using traditional methods (e.g., height and size) and wild populations.

## 2. Results

### 2.1. Genetic Diversity

Sixteen microsatellite loci were analyzed to assess the genetic diversity of four populations (Table 1). These microsatellite loci have shown high variability in previous studies and were considered suitable for analyzing the genetic structure of *C. gigas*. The values described below represent the average genetic diversity metrics across the populations.

The average number of alleles (*N_A_*) ranged from 5.50 to 8.50, observed heterozygosity (*H*_O_) ranged from 0.437 to 0.494, and expected heterozygosity (*H*_E_) ranged from 0.507 to 0.667. The GH, GW, and PD populations were out of Hardy–Weinberg equilibrium (*P*_HWE_), but the BR population was in equilibrium at *P*_HWE_. Inbreeding was observed in the GH, GW, and PD populations, with the inbreeding coefficient (*F*_IS_) being significant. The GH and PD populations, which were developed for resource creation, had *H*_O_ values of 0.473 and 0.455, respectively, showing higher genetic diversity than the GW (wild) population. The farmed BR population (from Japan) exhibited the highest level of genetic diversity with an *H*_O_ of 0.494.

### 2.2. Bottleneck Test and Effective Population Size Analysis

Under the infinite allele mutation model (IAM), we detected significant bottlenecks in all populations (*p* < 0.001). Bottlenecks were also identified in the BR population using the two-phase mutation model (TPM) and the stepwise mutation model (SMM) (Table 2). The BR population exhibited recent mode shifts, providing evidence of a bottleneck.

The effective population size across the four populations ranged from 82 to 678 (Table 2). The GW population had the largest effective population size at 678, while the GH population had the smallest at 82 (Table 2). The effective population sizes of the GH and PD populations, which were established for resource creation, were 82 and 403, respectively (Table 2). The GH population’s size was below the minimum effective population size of 100 required to prevent inbreeding depression.

### 2.3. Population Structure and Genetic Differentiation Analyses

Using microsatellite datasets, the pairwise genetic differentiation (*F*_ST_) values were found to be significant between BR and GH, GW, and PD populations, with the highest *F*_ST_ value observed between the BR and PD populations (*F*_ST_ = 0.048, Table 3). In contrast, the GH, GW, and PD populations exhibited very low *F*_ST_ values among each other (*F*_ST_ = 0.009–0.010).

Bayesian clustering analysis maximized the delta population constant (Δ*K*) value for population structure at *K* = 2 (Figure 1). The next maximum was the delta *K* (ΔK) value for the population structure at *K* = 5 (Figure 1). The STRUCTURE analysis results clearly separated the BR population from the GH, GW, and PD populations (Figure 1). At *K* = 2, the BR population formed the first group, while the GH, GW, and PD populations constituted the second group. Interestingly, the artificially derived GH and PD populations, originating from the BR population, were genotypically similar to the wild GW population.

Scatterplots from the discriminant analysis of principal components (DAPC) also indicated that the BR population formed the first group, while the GH, GW, and PD populations comprised the second group. These findings were consistent with the results from the STRUCTURE analysis (Figure 2).

## 3. Discussion

### 3.1. Genetic Diversity

Genetic diversity is an important factor for species to adapt to environmental changes [17]. Species-level genetic diversity can be used to determine the relative health of a species compared to other species [18]. The Chinese population (Li et al., 2006, China) showed a range of H_O_ of 0.474–0.616 [1], confirming that the genetic diversity of Korean and BR (Japanese broodstock) populations was lower than that of the Chinese population (H_O_ = 0.437–0.494). In this study, the BR population was a group created by importing Japanese broodstock. One of the reasons for importing the Japanese population was that the genetic diversity of the Japanese population was higher than that of the Korean population. The broodstock population was created for the purpose of resource creation, and a population with high H_O_, that is, high genetic diversity, was selected and created. Therefore, the BR population showed higher genetic diversity than the wild population GW, or the GH and PD populations for resource creation. This means that BR populations have higher genetic diversity than wild populations and can better withstand changes in the external environment. However, although the observed heterozygosity was high, the number of alleles was relatively small compared to the wild population GW.

This is one of the common problems in farmed populations [17,19,20]. This problem arises because it is difficult to secure a large number of individuals to create a parent population. These founder effects can create genetic differences by changing the allele frequencies between previous populations and current populations [6]. To address this issue, we propose creating a parent population with diverse alleles, rather than conducting stock enhancement alone.

Interestingly, the GW population has a high number of alleles despite the low observed heterozygosity. This could be due to the increase in alleles caused by the transplantation of a different population into the GW population. Although genetic diversity is low, the transplantation of a new population can increase the number of alleles. In fact, since the 1970s, fishermen have transplanted C. gigas from the southern coast of Korea to Cheonsu Bay in Chungnam Province several times [21] [local fisherman, personal communication]. This transplantation of a new population could have introduced new alleles into the population and increased the number of alleles. However, if a new population was introduced, the F_IS_ value should have decreased; the reason for the high F_IS_ value is thought to be as follows: in the case of the GW population, Cheonsu Bay, where GW is located, is a semi-closed bay with geographical characteristics that severely restrict water circulation (Figure 3). Even if individuals transplanted from the southern coast were incorporated into the GW population, the geographical characteristics likely caused individuals confined in the inner bay to have a high probability of inbreeding through mating over an extended period of time. Although the exact cause of the increased allelic diversity and elevated F_IS_ value observed in the GW population is unknown, historical records and geographical constraints suggest that sustained mating within a transplanted, closed population is a likely explanation.

Another possible cause is increased inbreeding resulting from population declines caused by oil spills. The Ganwoldo (GW) area was directly affected by the Hebei Spirit crude oil spill in December 2007 [22]. The oil spill has been reported to have reduced biodiversity in the intertidal area, where C. gigas lives [22]. Therefore, it can be inferred that the C. gigas population also decreased. Typically, population declines lead to bottlenecks, which increase the likelihood of inbreeding and reduce genetic diversity (H_O_) [23]. The low genetic diversity and high inbreeding, despite being a wild population, are thought to be due to the impact of the oil spill. This study showed that the F_IS_ values of GH and PD, which were created for the purpose of resource creation, were lower than that of the wild GW population. However, it did not show a significant decrease in the F_IS_ value, which can be explained by two hypotheses. The first hypothesis is that, although the maternal lineage was introduced from the BR population, inbreeding may have occurred within the wild population due to the low survival rate of the immigrant individuals or larvae released from them, resulting in limited genetic influence of the BR population. The second hypothesis is that since the area is an open sea rather than a closed space, the F_IS_ value may not have decreased significantly even if Korean and BR individuals were mixed, as the number of BR individuals was too small to have a noticeable effect. Based on the results, the latter is more likely. This is because if the number of broodstock is small, the probability of crossbreeding is low, making it difficult to increase the genetic diversity of the Korean population. Therefore, it suggests that more broodstock are needed to increase the genetic diversity of the Korean GH and PD populations.

The IAM model is suitable for estimating recent bottlenecks [24]. In this study, all populations were found to have experienced a bottleneck, and the BR population showed a clear bottleneck as indicated by a mode shift. The reasons for this bottleneck are as follows: the bottleneck may have been caused by the founder effect, artificial crossbreeding, or inbreeding. In the case of the BR population, it may have experienced a bottleneck because it imported Japanese broodstock and bred from this broodstock for more than five generations. The GW population is a wild population, and it may have experienced a bottleneck due to events such as overfishing, a large-scale population decline caused by summer mortality disease (high temperature), or the crude oil spill [22].

Ne is important for species adaptation and the maintenance of evolutionary potential in changing environments [24,25]. Populations with small effective population sizes are at an increased risk of local extinction due to genetic drift and inbreeding effects over generations [24,25]. In this study, populations with Ne sizes of less than 100, such as GH, are likely to experience a decrease in heterozygosity over short generations and a decrease in genetic diversity as homozygosity increases [6], which is supported by the significant F_IS_ value. The wild population, GW, had an effective population size of 678, and its genetic diversity is expected to gradually decrease if it falls below 1000 in the long term [6]. One reason for this is the high and significant F_IS_ value. However, because the effective population size is larger than that of other populations, the GW population is more likely to continue to thrive. Interestingly, the GH and PD populations, which were established for resource creation, show differences in effective population sizes. This can be attributed to two possibilities. The first possibility is that the GH population may have been influenced by the BR population. The second possibility is that the original oyster population that was not influenced by the BR population may have had a small effective population size. Both hypotheses are possible, but more population and genotypic analyses are needed to investigate this clearly.

To enhance the genetic diversity of the Korean GH and PD populations and prevent the loss of genetic variation due to inbreeding, we propose the following: (1) establish a genetically diverse maternal population through selective breeding, rear it to reproductive maturity, and transplant the population with the highest possible allelic diversity; and (2) release the population only after confirming, through genotypic analysis, that the effective population size is sufficiently large.

### 3.2. Genetic Structure

Differences in genetic structure are usually caused by reproductive isolation [26]. East Asian and North American Pacific oysters were considered a single group because their oceanic regions are connected [27]. However, previous studies have reported differences between Chinese and Korean, Japanese, and Canadian populations [27]. These differences are genetic statistically significant, and differences can often be found between farmed and wild populations [8]. In this study, genetic differences were confirmed between the Japanese broodstock population, BR, and GH, GW, and PD. Although the *F*_ST_ value did not show a high level of genetic differentiation, it indicated significant differentiation, so the results of genetic differences are consistent with the STRUCTURE results. The DAPC results also showed the same trend of genetic differences as the STRUCTURE results, supporting this conclusion. Previous studies have shown no genetic differences between Korea and Japan, but the genetic differences observed in this study are thought to be due to the following reasons [27]: Japanese broodstock have been bred for more than five generations, and indices such as allele frequency may differ from those of the wild population. This is due to the founder effect, where a small number of individuals from the original population show differences in allele frequencies as generations pass [28]. For this reason, it is concluded that there is a genetic statistically significant difference between the BR population and the wild population. This can be found in previous studies, where results showing genetic differences between farmed and wild populations are documented [8].

When looking at the genetic differences, they were significant, and since *H*_O_, which is an index indicating genetic diversity, is high, it is considered useful for increasing genetic diversity by using it as broodstock. However, since the number of alleles is insufficient, it is proposed to increase the genetic diversity of oysters in the future by reducing inbreeding and supplementing the number of alleles through crossbreeding between the Korean and Japanese populations.

## 4. Materials and Methods

### 4.1. Sampling and DNA Extraction

*Crassostrea gigas* is used as an inland fishery resource for commercial fishing and aquaculture in Korea, and is exempt from animal ethics approval because its suffering level is classified as A. In this study, the BR population was a group created by importing Japanese broodstock. The BR population was imported from Hiroshima (Japan) in August 2018. One of the reasons for importing the Japanese population was that the genetic diversity of the Japanese population was higher than that of the Korean population. The broodstock population was created for the purpose of resource creation, and a population with high *H*_O_, that is, high genetic diversity, was selected and created. The BR group is a group selected through existing traditional breeding methods (size) and bred continuously in the Tongyeong waters for more than five generations. One of the reasons for using the BR population for resource creation was that it had high genetic diversity, so the goal was to improve genetic diversity through crossbreeding with the population being farmed. The PD and GH populations of *C. gigas* were released from the BR population, so the farmed population (BR) was used as the sample. We also sampled populations using resource generation populations (GH, PD) and GW populations. Overall four populations of *C. gigas* were sampled (location and latitude–longitude details are provided in Figure 4 and Table 4). Each sampling was conducted in 2023, and mantle tissue from the Pacific oysters were collected and soaked in 99% ethanol. Genomic DNA (gDNA) was extracted using the DNeasy Blood & Tissue Kit (QIAGEN, Hilden, Germany) according to the manufacturer’s instructions. The gDNA extracted was diluted to a concentration of 50 ng/μL and subsequently stored at −20 °C for the purpose of amplifying microsatellite loci.

### 4.2. Microsatellite Genotyping

We selected sixteen microsatellite loci developed in previous studies and performed PCR using an Applied Biosystems™ ProFlex™ PCR System (Thermo Fisher Scientific, Foster City, CA, USA) [29,30]. PCR was performed using 50 ng of gDNA, 0.5 units of AccuPower^®^ PCR PreMix (Bioneer, Daejeon, Republic of Korea), 0.5 μM of forward primer labeled with fluorescent dyes (FAM, HEX, TAMRA, and ATTO565), and 0.5 μM of reverse primer. The PCR conditions were as follows: pre-denaturation at 95 °C for 5 min, denaturation at 95 °C for 20 s, annealing at 60 °C for 20 s, and extension at 72 °C for 20 s. After 35 cycles, final extension was performed at 72 °C for 10 min, and the temperature was then held at 8 °C. Each amplified PCR product was electrophoresed on a 2% agarose gel to confirm the presence or absence and the size of the amplified fragment. The PCR fragments were prepared by mixing them with a GeneScan™ 500 ROX size standard ladder (Thermo Fisher Scientific) and HiDi™ formamide, followed by denaturation at 95 °C for 2 min and termination at 4 °C. The raw data of allele sizes were determined using an Applied Biosystems™ ABI 3730xl DNA Analyzer (Thermo Fisher Scientific). Genotyping was performed using GeneMapper software (version 5) [31].

### 4.3. Genetic Diversity Analysis

Scoring errors in microsatellite loci were assessed using MICRO-CHECKER software (version 2.2.3) [32]. Genetic diversity was quantified by calculating the number of alleles (*N*_A_), expected heterozygosity (*H*_E_), and observed heterozygosity (*H*_O_) using CERVUS software (version 3.0) [33]. The inbreeding coefficient (*F*_IS_) and tests for Hardy–Weinberg equilibrium (*P*_HWE_) were conducted using GENEPOP (version 4.2) [34] and ARLEQUIN software (version 3.5) [35]. To estimate population bottlenecks, two methods were employed. The first involved the BOTTLENECK software (version 1.2.02) [36], which estimates bottlenecks through tests for heterozygote excess under the infinite alleles mutation model (IAM) [37]. The second method involved a two-phase mutation model (TPM) and a stepwise mutation model (SMM) [37], with TPM implemented using 10% variance and 90% SMM. Both models were run with 10,000 iterations, and significance was assessed using the Wilcoxon signed-rank test [38]. The effective population size (Ne) was estimated using the linkage disequilibrium method (Parametric CI) with NeEstimator software (version 2.1) [39]. Additionally, the effective population size was estimated with a 95% confidence interval.

### 4.4. Population Genetic Structure Analysis

STRUCTURE software (version 2.3) [40] was used to conduct a genetic structure clustering analysis based on a Bayesian method model. To determine the most appropriate number of populations (*K*), we tested values from 1 to 10, applying a suitable admixture model to account for the mixed water systems. The burn-in period was set to 10,000 iterations, followed by 100,000 iterations of Markov chain Monte Carlo, with the entire process repeated 10 times. To determine the optimal number of clusters, we analyzed the results corresponding to each *K* value using STRUCTURE SELECTOR (beta version) [41]. Additionally, a discriminant analysis of principal components (DAPC), a non-model-based genetic clustering method, was performed on the microsatellite dataset using the R package ADEGENET (version 2.1.3) [42].

## 5. Conclusions

Our aim was to understand the genetic diversity and structure of farmed and wild populations and to provide reference data for effective resource development strategies. The observed heterozygosity *H*_O_ in the farmed population was 0.494, while in wild and resource creation populations, it ranged up to 0.473. The farmed BR population exhibited the highest genetic diversity but had a low effective population size of 105. The PD population size, intended for resource creation, was 403, which was relatively high compared to the GH population. To supplement the population size, the introduction of wild broodstock is necessary. Genetic structure analysis divided the populations into two groups with *K* = 2: the first group comprised the BR population, while the second group included the GH, GW, and PD populations. This indicated significant genetic differences between the farmed population and both the wild and resource-building populations. This study underscores the need for breeding strategies to increase genetic diversity in Korean *C. gigas* aquaculture and suggests that such strategies may help maintain the genetic health of wild populations.

## Figures and Tables

**Figure 1 ijms-26-04172-f001:**
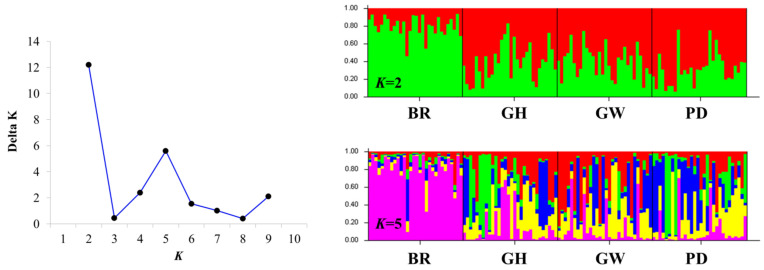
Population genetic structure of *C*. *gigas* (*K* = 2). The BR populations are farmed populations and the remaining populations are wild and resource generation populations. The graph displays the relevant delta *K* information for population constants. Each histogram represents the probability that an object, associated with a specific color, is assigned to a particular cluster.

**Figure 2 ijms-26-04172-f002:**
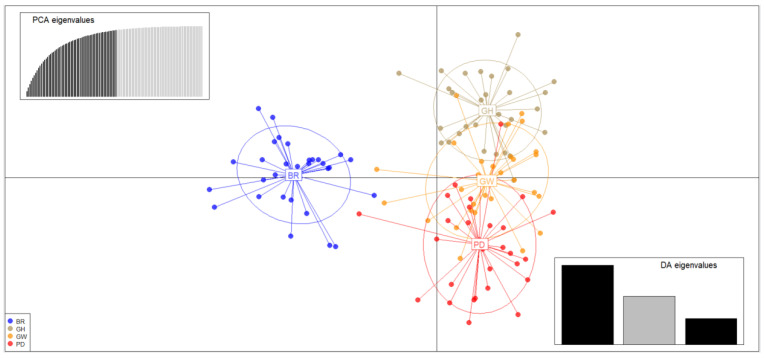
Scatterplots of discriminant analysis of principal components (DAPC) for *C. gigas*. In the plot, each color corresponds to a distinct population, representing different genetic clusters, with population abbreviations provided for each cluster. The graph in the upper left illustrates the contribution of the eigenvalues from the selected principal components, while the graph in the bottom right shows the variance explained by the eigenvalues of the two discriminant functions in the scatterplot.

**Figure 3 ijms-26-04172-f003:**
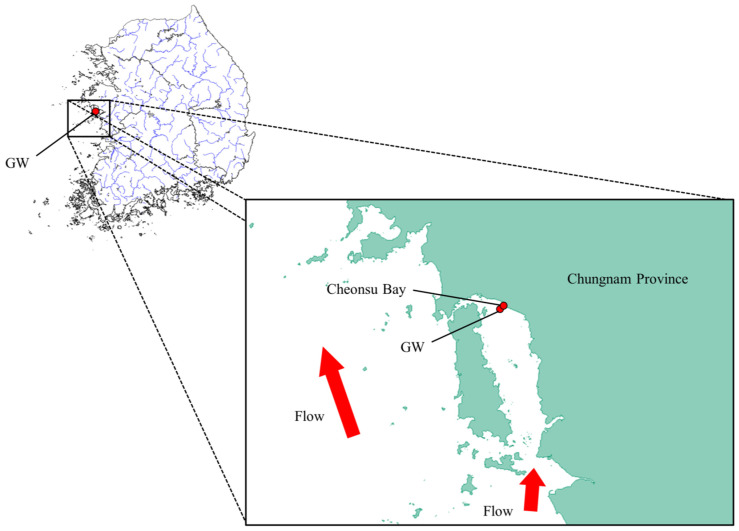
Geographical map of the area inhabited by GW populations.

**Figure 4 ijms-26-04172-f004:**
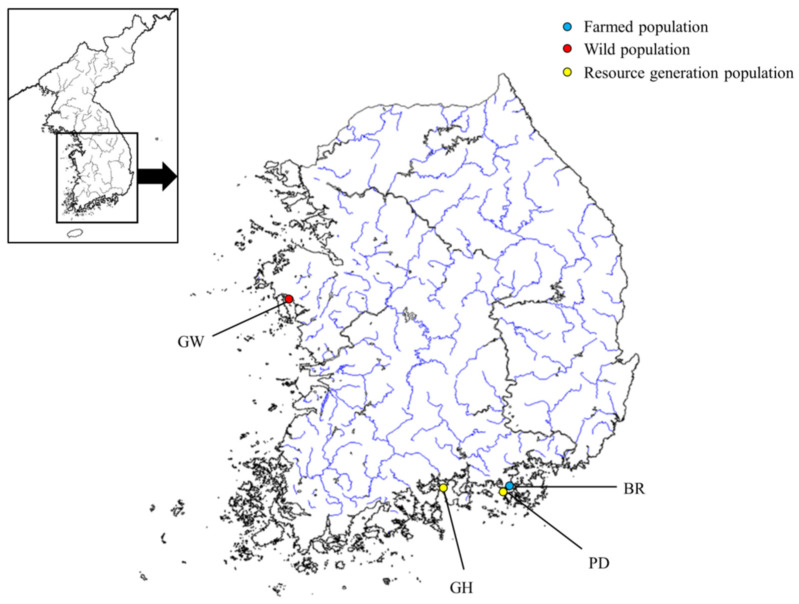
Sampling location of the four populations of *C. gigas*. Abbreviations of populations are given in Table 4. The BR population was sampled from a population imported from Japan (Hiroshima) in 2018 and selectively bred in Tongyeong waters for more than five generations.

**Table 1 ijms-26-04172-t001:** Summary information of genetic diversity for populations based on 16 microsatellite loci of *C. gigas*.

ID	Location	*N*	*N_A_*	*H* _O_	*H* _E_	*P* _HWE_	*F* _IS_
BR	farmed population (Japan, Hiroshima)	30	5.50	0.494	0.507	0.219	0.022
GH	Galhwa	30	7.88	0.473	0.621	0.000 ***	0.242 ***
GW	Ganwoldo Island	30	8.50	0.437	0.604	0.000 ***	0.280 ***
PD	Pildo	30	8.44	0.455	0.667	0.000 ***	0.269 ***

*N*: Number of samples, *N*_A_: Average number of alleles, *H*_O_: Observed heterozygosity, *H*_E_: Expected heterozygosity, *P*_HWE_: Hardy–Weinberg equilibrium value, *F*_IS_: Inbreeding coefficient, *** *p* < 0.001.

**Table 2 ijms-26-04172-t002:** Summary information of bottleneck test and effective population size for four populations.

Population ID	*N*	Wilcoxon Signed-Rank Test				*N* _e_	(95% CI)
		*P* _IAM_	*P* _TPM_	*P* _SMM_	Mode-Shift		
BR	30	0.000 ***	0.000 ***	0.011 *	Shifted	105	(47–∞)
GH	30	0.000 ***	0.068	0.135	L-shaped	82	(48–223)
GW	30	0.000 ***	0.532	0.978	L-shaped	678	(104–∞)
PD	30	0.000 ***	0.104	0.208	L-shaped	403	(99–∞)

*N*: Number of samples, *P*_IAM_: *p*-value of bottleneck test using infinite allele mutation model, *P*_TPM_: *p*-value of bottleneck test using two-phase mutation model (10% variance and 90% proportions of SMM), *P*_SMM_: *p*-value of bottleneck test using stepwise mutation model, *N*_e_: estimated effective population size via NeEstimator ver 1.3 software, CI: confidence interval, * *p* < 0.05, *** *p* < 0.001.

**Table 3 ijms-26-04172-t003:** *F*_ST_ among populations of *C*. *gigas* by microsatellite loci data.

	BR	GH	GW	PD
BR	-	0.000	0.000	0.000
GH	0.044 *	-	0.087	0.516
GW	0.040 *	0.010	-	0.184
PD	0.048 *	0.004	0.009	-

Pairwise genetic differentiation significant level (above), *F*_st_: Pairwise genetic differentiation (below), *: statistically significant *p* < 0.001.

**Table 4 ijms-26-04172-t004:** Details of the location information for *C*. *gigas*.

Location Information	Code	Location Name	*N*	Location
Farmed *	BR	farmed	30	34°47′30′′ N 128°25′46′′ E
Resource generation population ^1^	GH	Galhwa	30	34°53′57′′ N 127°50′06′′ E
Resource generation population ^1^	PD	Pildo	30	34°51′11′′ N 128°20′39′′ E
Wild population	GW	Ganwoldo Island	30	36°36′17′′ N 126°24′37′′ E

*N*: Number of samples, *: This population was imported from Japan (Hiroshima) in 2018 and selected and reared in Tongyeong for more than five generations, ^1^: This population was transplanted for resource creation from individuals selectively bred in BR (Hiroshima, Japan).

## Data Availability

The raw data supporting the conclusions of this article will be made available by the authors, without undue reservation.

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
