# Peer review of "Genetic Diversity and Population Structure in Farmed and Wild Pacific Oysters (Crassostrea gigas): A Comparative Study"

_ijms, 2025, doi:10.3390/ijms26094172_

Round 1
Reviewer 1 Report
Comments and Suggestions for Authors
The manuscript "Genetic Diversity and population structure in farmed and wild pacific oysters (Crassostrea gigas): a comparative study", evaluated the genetic diversity and population structure of three Korean populations compared with a farmed Japanese broodstock based on microsatellite loci.
After reviewing the manuscript, some points described below deserve attention and need to be reviewed carefully:
Lines 61-65: Provide the year the Japanese broodstock was imported, how you assigned each generation, and when the adapted broodstock was released. This temporal information can significantly contribute to the interpretation of the results discussed in the manuscript.
Lines 77-79: In Table 1, the Phwe was significant for GH, GW, and PD, and not to BR as you said. Fix it.
Lines 96-98: You must consider the Ne plus 95% CI, which was below the minimum expected (100) for all populations evaluated. Fix it.
Table 3: Provide the P-value significance used.
Figure 1: Why did you exhibit K=5 if you did not mention it in the results? If you maintain the k=5 graph, include details in the caption like what would be the different colors shared among the populations.
Lines 148-150: You must use caution "healthier" here because the BR population suffered a bottleneck event and is marginally below the minimum Ne=100 expected (Ne=105). Just because it has a different allele composition and frequency doesn't mean it's healthier or worse than the others.
Lines 151-154: This paragraph seems disconnected. I suggest transferring it next to the last paragraph of this subsection (lines 219-224).
Figure 3 + 4: I suggest joining figures 3 and 4 for a better spatial perspective, which also could include the main currents that flow among the BR, PD, and GH populations. This information could be relevant to evaluate how the larvae are carried with the current flow, which could explain the admixture levels among GH and PD populations, for example.
Line 263: The indication of Table 5 is not correct, indeed is Table 4. Fix it.
Lines 279-280: You changed the terms here. First, you genotype the microsatellite in the sequencer, then you score the size of the alleles in the GeneMapper. Fix it.
Lines 293-294: What confidence interval method was used, and What minimum errors were considered?
Lines 306-324: The conclusion seems more abstract than an overall perspective of the manuscript findings. I recommend reformulating it by providing, for example, How your study could contribute to improving the genetic rescue of farmed populations to achieve the wild population Ne=678 and Ar=8.50 levels.
Author Response
For research article
Response to Reviewer 1 Comments
|
||
1. Summary |
|
|
Thank you very much for taking the time to review this manuscript. Please find the detailed responses below and the corresponding revisions/corrections highlighted/in track changes in the re-submitted files.
|
||
|
|
|
|
|
|
|
|
|
|
|
|
|
|
|
|
|
|
|
|
|
2. Point-by-point response to Comments and Suggestions for Authors |
||
Comments 1: Lines 61-65: Provide the year the Japanese broodstock was imported, how you assigned each generation, and when the adapted broodstock was released. This temporal information can significantly contribute to the interpretation of the results discussed in the manuscript.
|
||
Response 1: Thank you for pointing this out. [In this study, the BR population was a group created by importing Japanese broodstock. The BR population was imported from Hiroshima (Japan) in August 2018. One of the reasons for importing the Japanese population was that the genetic diversity of the Japanese population was higher than that of the Korean population. The broodstock population was created for the purpose of resource creation, and the population with high HO, that is, high genetic diversity, was selected and created. The BR group is a group selected through existing traditional breeding methods (size) and bred continuously in the Tongyeong waters for more than five generations. One of the reasons for using the BR population for resource creation was that it had high genetic diversity, so the goal was to improve genetic diversity through crossbreeding with the population being farmed.]
|
||
Comments 2: Lines 77-79: In Table 1, the Phwe was significant for GH, GW, and PD, and not to BR as you said. Fix it. Response 2: Thank you for your review. [Hardy-Weinberg equilibrium (PHWE), but the BR population was in equilibrium at PHWE.] |
Comments 3:
Table 3: Provide the P-value significance used.
Response 3: Thank you for your review.
[*: statistically significant P<0.001.]
Comments 4:
Figure 1: Why did you exhibit K=5 if you did not mention it in the results? If you maintain the k=5 graph, include details in the caption like what would be the different colors shared among the populations.
Response 4: Thank you for your review.
K=5 is mentioned.
Same color indicates probability of clustering together.
[The next maximum is the delta K (ΔK) value for the population structure at K = 5 (Figure 1).]
[Each histogram represents the probability that an object, associated with a specific color, is assigned to a particular cluster.]
Comments 5:
Lines 148-150: You must use caution "healthier" here because the BR population suffered a bottleneck event and is marginally below the minimum Ne=100 expected (Ne=105). Just because it has a different allele composition and frequency doesn't mean it's healthier or worse than the others.
Response 5:
Thank you for your review.
[This means that BR populations have higher genetic diversity than wild populations and can better withstand changes in the external environment.]
Comments 6:
Lines 151-154: This paragraph seems disconnected. I suggest transferring it next to the last paragraph of this subsection (lines 219-224).
Response 6:
Thank you for your review. Revised as suggested by reviewer 2 to strengthen the coherence of paragraphs.
[This is one of the common problems in farmed populations [17,19,20]. This problem arises because it is difficult to secure a large number of individuals to create a parent population. These founder effects can create genetic differences by changing the allele frequencies between previous populations and current populations [6]. To solve this problem, rather than simply reinforcing the population, we propose creating a parent population with diverse alleles.]
Comments 7:
Figure 3 + 4: I suggest joining figures 3 and 4 for a better spatial perspective, which also could include the main currents that flow among the BR, PD, and GH populations. This information could be relevant to evaluate how the larvae are carried with the current flow, which could explain the admixture levels among GH and PD populations, for example.
Response 7:
Thank you for your review. The map in Figure 4 is a point about sampling. If combined with Figure 3, it will not be easy to explain and understand in terms of the flow of context. Therefore, please understand the point of dividing it into its current state.
Comments 8:
Line 263: The indication of Table 5 is not correct, indeed is Table 4. Fix it.
Response 8:
Thank you for your review. The manuscript has been revised.
Comments 9:
Lines 279-280: You changed the terms here. First, you genotype the microsatellite in the sequencer, then you score the size of the alleles in the GeneMapper. Fix it.
Response 9:
Thank you for your review. The raw data of the allele size was corrected. After that, the genotype was determined.
[The raw data of allele sizes were determined using an Applied Biosystems™ ABI 3730xl DNA Analyzer (Thermo Fisher Scientific). Genotyping was performed using GeneMapper software (version 5) [31].]
Comments 10:
Lines 293-294: What confidence interval method was used, and What minimum errors were considered?
Response 10:
Thank you for your review.
[The effective population size (Ne) was estimated using the linkage disequilibrium method (Parametric CI) with NeEstimator software (version 2.1) [39]. Additionally, the effective population size was estimated with a 95% confidence interval.]
Comments 11:
Response 11:
Thank you for your review. The conclusions have been reconstructed based on the comments of reviewers 1 and 2.
[Our aim was to understand the genetic diversity and structure of farmed and wild populations and to provide reference data for effective resource development strategies. The observed heterozygosity in farmed populations HO ranged from 0.494, while in wild populations, it ranged from 0.473. The farmed BR population exhibited the highest genetic diversity but had a low effective population size of 105. The PD population size, intended for resource creation, was 403, which was relatively high compared to the GH population. To supplement the population size, the introduction of wild broodstock is necessary. The genetic structure analysis divided the populations into two groups with K = 2: the first group comprised the BR population, while the second group included the GH, GW, and PD populations. This indicated significant genetic differences between the farmed populations and both the wild and resource-building populations. This study underscores the need for breeding strategies to increase genetic diversity in Korean C. gigas aquaculture and suggests that such strategies may help maintain the genetic health of wild populations.]

Reviewer 2 Report
Comments and Suggestions for Authors
Please see all my comments in the text. There should be description of the localities/populations. Neither Japan nor Chinese populations have been studied, but are commented. Nothin has been said about the procedures of specimen collection/sampling design. Why, instead of sampling in the Japanese population some specimens, rpelaced to Korea, were the source of "Japan population"? The ms is not easy to be followed. Some of the results and conclusions are not well supported and clearly presented or obvious without any study. The genetic data seems interesting, but neither their interpretation nor the scheme of the study cannot be accepted in their present form, in my opinion.

In general, the English is not bad, but several statements should be rephrased, and the text is not always clear.
Author Response
For research article
Response to Reviewer 2 Comments
|
||
1. Summary |
|
|
Thank you very much for taking the time to review this manuscript. Please find the detailed responses below and the corresponding revisions/corrections highlighted/in track changes in the re-submitted files.
|
||
|
|
|
|
|
|
|
|
|
|
|
|
|
|
|
|
|
|
|
|
|
2. Point-by-point response to Comments and Suggestions for Authors |
||
Comments 1: Please see all my comments in the text. There should be description of the localities/populations. Neither Japan nor Chinese populations have been studied, but are commented. Nothin has been said about the procedures of specimen collection/sampling design. Why, instead of sampling in the Japanese population some specimens, rpelaced to Korea, were the source of "Japan population"? The ms is not easy to be followed. Some of the results and conclusions are not well supported and clearly presented or obvious without any study. The genetic data seems interesting, but neither their interpretation nor the scheme of the study cannot be accepted in their present form, in my opinion.
|
||
Response 1: Thank you for pointing this out. The main point of the paper in this study is that there is a difference in genetic structure between farmed and wild oysters. That is, the oysters that have been farmed for more than 4 generations are bred through traditional selective breeding (selecting large-sized individuals), etc., so the allele frequencies in the wild and the resource-creating populations will differ, resulting in a difference in genetic structure between the farmed population (Japan) and the rest of the wild and resource-creating populations.
|
||
Comments 2: from 0.494 to what value? Response 2: Thank you for your review. [The observed heterozygosity in the farmed population HO ranged from 0.494, while in the wild population, it was 0.473.] |
Comments 3:
all these keywords are already in the title, thus should be replaced with some other ones
Response 3: Thank you for your review.
[microsatellite; effective population size; aquatic animals; shellfish]
Comments 4:
author, date
Response 4: Thank you for your review.
[Wild Pacific oysters (Crassostrea gigas; An et al. 2013, Korea; Li et al. 2006, China; Zhang et al. 2023, China) from China and Korea are reported to have low genetic diversity [1-3].]
Comments 5:
but this is a Japanese, possibly endemic species, thus how its genetic diversity in Europe may be higher>
Response 5:
Thank you for your review.
[This species is not endemic to C. gigas. Research suggests that genetic differences are due to changes in allele frequency caused by the founder effect.]
Comments 6:
where?
Response 6:
Thank you for your review.
[Since Pacific oysters are also reported to have low genetic diversity [3], attempts to reinforce them using the same method as in the previous study are needed to increase genetic diversity and effective population size [8].]
Comments 7:
somewhat unclear and should be transferred to material and methods
Response 7:
Thank you for your review.
It has been removed from the manuscript.
Comments 8:
how there could be a half of allele? Perhaps you should explain the mean number?
Response 8:
Thank you for your review.
[The average number of alleles (NA) ranged from 5.50 to 8.50, observed heterozygosity (HO) ranged from 0.437 to 0.494, and expected heterozygosity (HE) ranged from 0.507 to 0.667.]
Comments 9:
FIS does not simply quantify the inbreeding
Response 9:
Thank you for your review.
FIS does not simply quantify inbreeding, but rather allows for comparison of relative levels of inbreeding. Thank you for your understanding.
Comments 10:
bur BR is in Korea at your map
Response 10:
Thank you for your review.
[The BR population was sampled from a population imported from Japan (Hiroshima) in 2018 and selectively bred in Tongyeong waters for more than five generations.]
Comments 11 :
all this paragraph, rephrased and written in a more clear way, should be transferred to material and methods, and perhaps to introduction.
Response 11:
Thank you for your review. The BR group is described in Materials and Methods.
[In this study, the BR population was a group created by im-porting Japanese broodstock. One of the reasons for importing the Japanese population was that the genetic diversity of the Japanese population was higher than that of the Korean population. The broodstock population was created for the purpose of resource creation, and the population with high HO, that is, high genetic diversity, was selected and created. The BR group is a group selected through existing traditional breeding methods (size) and bred continuously in the Tongyeong waters for more than five generations. One of the reasons for using the BR population for resource creation was that it had high genetic diversity, so the goal was to improve genetic diversity through crossbreeding with the population being farmed.]
Comments 12:
which one? All the four studied populations are in Korea.
again, Japan appears in one table, but not in the map
Response 12:
Thank you for your review. China mentioned the author. It is a Japanese group, but is currently located in the Tongyeong area.
[The Chinese population (Li et al. 2006, China) showed a range of HO of 0.474-0.616 [1], confirming that the genetic diversity of Korean and BR (Japanese broodstock) populations was lower than that of the Chinese population (HO = 0.437-0.494).]
Comments 13:
this should be stated clearly, and much earlier in the text, and necessarily in the methods; and such "created" population may differ from the source one. Why you have not studied specimens taken directly from the Japanes population?
Response 13 :
Thank you for your review.
This is because we are not comparing the wild Japanese population with the Korean wild population. The Japanese population had a high genetic diversity and there was a legal basis for releasing them only after a certain number of generations of farming had been completed, so the wild Japanese population was not used. So the reason why the wild Japanese population was not used was for legal reasons and the fact that genetic diversity was maintained.
Comments 14:
I can see only one population
Response 14:
Thank you for your review.
GW, GH, and PD are the same population.
[Therefore, the BR population showed higher genetic diversity than the wild population GW, GH and PD populations for resource creation.]
Comments 15:
absolutely unclear for me, I have positively lost the thread as concerns populations.
Response 15:
Thank you for your review.
Genetic population structure is BR vs. GW, GH, PD.
Comments 16:
this may be simply interpreted as founder effect: introducing a species the number of specimens is usually low, the source populations were chosen without any knowledge of the genetics, thus it is just a clear case of founder effect, in this case a man-made one
Response 16:
Thank you for your review.
[These founder effects can create genetic differences by changing the allele frequencies between previous populations and current populations [6].]
Comments 17:
as it is clearly seen in the table, the heterosigosity was the same as in all the other populations. Considering the higher number of alleles, the expected heterozygosity should be higher, but it is another story
Response 17:
Thank you for your review. As can be clearly seen from the table, heterozygosity is not the same for all other groups, which shows relatively low HO through MS markers.
Comments 18:
trnsplantation of all the population? Not some specimens?
Response 18:
Thank you for your review.
[Some sample sizes (e.g., 100 individuals, where the effect is minimal) were difficult for populations to thrive, and it is assumed that population transplants at that time resulted in large numbers of C. gigas being transplanted.]
Comments 19:
unusual word in this context
Response 19:
Thank you for your review.
[Even if individuals transplanted from the southern coast were incorporated into the GW population, the geographical characteristics likely caused individuals confined in the inner bay to have a high probability of inbreeding through mating over an extended period of time.]
Comments 20:
a transplantation as such per se cannot be the reason of inbreeding - nothin more than unsuccessful decrease of inbreeding may be result of transplantation.
Response 20:
Thank you for your review.
[Although the exact cause of the increased allelic diversity and elevated FIS value observed in the GW population is unknown, historical records and geographical constraints suggest that sustained mating within a transplanted, closed population is a likely explanation.]
Comments 21:
what does it mean?
Response 21:
Thank you for your review. It has been deleted.
Comments 22:
why such a map is not in Material and Methods? It should not be given in Discussion;
Again, why there is no any description of the localities im material and methods?
Response 22:
Thank you for your review. The reason that this map is not included is because the map in Materials and Methods is a map of the entire sample. It is redundant and difficult to read if it is explained in Materials and Methods, so it is shown separately. Thank you for your understanding.
Comments 23:
in which? Numbers? Genetic diversity? or which else?
Response 23 :
Thank you for your review.
[Typically, population declines lead to bottlenecks, which increase the likelihood of inbreeding and reduce genetic diversity (HO)]
Comments 24:
somewhat more general note:
Why have you chosen, for comparisons, a "wild" population inhabiting such curious and heavily man-affected place?
Response 24:
Thank you for your review. Among the wild populations of C. gigas inhabiting the Korean Peninsula, we selected a population inhabiting the northwestern part of the west. At the time of collection, the population was selected from a place less affected by humans, and the population history of this population was later revealed through research.
Comments 25:
survival of what? Immigrants?
impact of population?
Response 25 :
Thank you for your review.
[The first hypothesis is that, although the maternal lineage was introduced from the BR population, inbreeding may have occurred within the wild population due to the low survival rate of the immigrant individuals or larvae released from them, resulting in a limited genetic influence of the BR population.]
Comments 26:
obvious without any study
Response 26:
Thank you for your review. It has been deleted.
Comments 27:
In general, Crassostrea has a pelagic phase. Nothing about this has been said anywhere, but this is important for the genetic structure
Response 27:
Thank you for your review. In general, C. gigas goes through a floating stage, and although these floating larvae sometimes travel with the flow of the current, it is thought that it is difficult to spread more widely in the closed side. It is thought that the oysters living in the intertidal water area at the time of actual collection were widely spread in the intertidal zone, but they exist in groups concentrated only in some parts of the intertidal zone in the collection area. Therefore, this reason may affect the genetic structure.
Comments 28:
is a population an importer?
Response 28:
Thank you for your review. Here, "it" is the BR population.
Comments 29:
delete
Response 29:
Thank you for your review. It has been removed from the manuscript.
Comments 30:
??
Response 30:
Thank you for your review.
[summer mortality disease (high temperature),]
Comments 31:
eggs are produced independently of a population size
Response 31:
Thank you for your review.
[To enhance the genetic diversity of the Korean GH and PD populations and prevent the loss of genetic variation due to inbreeding, we propose the following: 1) establish a genetically diverse maternal population through selective breeding, rear it to reproductive maturity, and transplant the population with the highest possible allelic diversity; and 2) release the population only after confirming, through genotypic analysis, that the effective population size is sufficiently large.]
Comments 32:
and what about selection, mutations?
Response 32 :
Thank you for your review. Mutations are thought to have little effect in changing the allele frequencies that affect a population.
Comments 33:
not East Asia
Response 33:
Thank you for your review. East Asian and North American Pacific oysters were considered a single group because their oceanic regions are connected [27].
Comments 34:
statistically?
Response 34:
Thank you for your review.
[These differences are genetic statistically significant, and differences can often be found between farmed and wild populations [8].]
Comments 35:
what does it mean?
Response 35:
Thank you for your review.
[For this reason, it is concluded that there is a genetic statistically significant difference between the BR population and the wild population.]
Comments 36:
In my opinion, all the localities should be described, and, which is more important, a clear scheme of the sampling and manipulations (introductions, numbers of introduced specimens, time of waiting for the results, etc., should be presented. Without this it is not easy to understand ALL the study
Response36 :
Thank you for your review. A clear plan for sampling and manipulation (introduction, number of samples introduced, waiting time for results, etc.) was described.
Comments 37:
bivalve as a fish? I do not understand
Response 37:
Thank you for your review.
[Crassostrea gigas is used as an inland fishery resource for commercial fishing and aquaculture in Korea, and is exempt from animal ethics approval because its suffering level is classified as A.]
Comments 38:
frankly, I cannot understand which population has been the source of specimens (GH, PD?), which the wild (for comparison), and which has been this studied after introduction of immigrants?
Response 38:
Thank you for your review.
[]
Comments 39:
how a population can be released from population?
again, only one
there is no Table 5
Response 39:
Thank you for your review. This has been reflected in the revised manuscript.
Comments 40:
these two columns are unnecessary, since the data for each population are the same - instead, they should be given in the caption of the table. Instead, some characteristic of the locality should be given.
Response 40:
Thank you for your review.
[*: This population was imported from Japan (Hiroshima) in 2018 and selected and reared in Tongyeong for more than five generations, 1: This population was transplanted for resource creation from individuals selectively bred in BR (Hiroshima, Japan).]
Comments 41:
it has already been said above
Response 41:
Thank you for your review. Deleted from the manuscript.
Comments 42:
obvious and generally known - should you provide some new, npteworthy results in Conclusions?
Response 42:
Thank you for your review. Deleted from the manuscript.

Reviewer 3 Report
Comments and Suggestions for Authors
The paper is studying and comparing genetic diversity and population structure in armed and wild pacific oysters (Crassostrea gigas) in Korea. Significant genetic differences between farmed, wild and resource creation populations were found.
The introduction is short and to the point. I do question the term "reinforcement method". Production of seed for "fisheries" has other terms, such as stock enhancement, stock reinforcement etc. The problems regarding this strategy is well described in the discussion, line 151-154.
I have no more specific comments.
Author Response
For research article
Response to Reviewer 3 Comments
|
||
1. Summary |
|
|
Thank you very much for taking the time to review this manuscript. Please find the detailed responses below and the corresponding revisions/corrections highlighted/in track changes in the re-submitted files.
|
||
|
|
|
|
|
|
|
|
|
|
|
|
|
|
|
|
|
|
|
|
|
2. Point-by-point response to Comments and Suggestions for Authors |
||
Comments 1: The introduction is short and to the point. I do question the term "reinforcement method". Production of seed for "fisheries" has other terms, such as stock enhancement, stock reinforcement etc. The problems regarding this strategy is well described in the discussion, line 151-154.
|
||
Response 1: Thank you for pointing this out.
|
[This is one of the common problems in farmed populations [17,19,20]. This problem arises because it is difficult to secure a large number of individuals to create a parent population. These founder effects can create genetic differences by changing the allele frequencies between previous populations and current populations [6]. To address this issue, we propose creating a parent population with diverse alleles, rather than conducting stock enhancement alone.]

Round 2
Reviewer 2 Report
Comments and Suggestions for Authors
Congratulations, I do hope that the ms is now much better.
